# Cage-confined photocatalysis for wide-scope unusually selective [2 + 2] cycloaddition through visible-light triplet sensitization

Jing-Si Wang[1], Kai Wu [1], Changzhen Yin[1], Kang Li[1], Yahao Huang[1], Jia Ruan[1], Ximin Feng[1], Peng Hu [1✉] & Cheng-Yong Su [1✉]

Light-induced [2 + 2] cycloaddition is the most straightforward way to generate cyclobutanes, which are core structures of many natural products, drugs and bioactive compounds. Despite continuous advances in selective [2 + 2] cycloaddition research, general method for intermolecular photocatalysis of acyclic olefins with specific regio- and diastereoselectivity, for example, syn-head-to-head (syn-HH) cyclobutane derivatives, is still lack of development but highly desired. Herein, we report a cage-confined photocatalytic protocol to enable unusual intermolecular [2 + 2] cycloaddition for $\alpha,\beta$-unsaturated carbonyl compounds. The syn-HH diastereomers are readily generated with diastereoselectivity up to 99%. The cage-catalyst is highly efficient and robust, covering a diverse substrate range with excellent substituent tolerance. The mimic-enzyme catalysis is proposed through a host-guest mediated procedure expedited by aqueous phase transition of reactant and product, where the supramolecular cage effect plays an important role to facilitate substrates inclusion and pre-orientation, offering a promising avenue for general and eco-friendly cycloaddition photocatalysis with special diastereoselectivity.

[1] MOE Laboratory of Bioinorganic and Synthetic Chemistry, Lehn Institute of Functional Materials, School of Chemistry, Sun Yat-Sen University, 510275 Guangzhou, China. ✉email: hupeng8@mail.sysu.edu.cn; cesscy@mail.sysu.edu.cn

Cyclobutane scaffolds are prevalent in natural products and bioactive compounds[1–3], therefore triggering tremendous synthetic interests in constructing four-membered carbocycles[4,5]. Since the first report in 1877[6], UV light-induced [2 + 2] cycloaddition of alkenes has been conveniently used to generate cyclobutane backbones[7]. However, geometric isomerization and racemic background photoreactions facilely induced by UV light make the stereochemical control of cycloaddition a long-standing challenge, especially for acyclic alkenes[8,9]. As a green and clean procedure for chemical process and production that are contemporarily demanded by sustainable development, the visible-light sensitized intra- and intermolecular [2 + 2] cycloaddition of olefins has drawn great attention of chemical society in past decades[10–15], leading to fast progress in [2 + 2] photocycloaddition of nonrigid olefins with excellent regio- and stereoselectivities[16–20]. Nevertheless, owing to steric preference and relative radical stability of excited-state intermediates, hitherto, the stereochemical control is mainly achieved for the thermodynamic favored anti-diastereomers from acyclic olefins, e.g., styrenes, chalcones, and cinnamates[16,21–24]. A general synthetic strategy for thermodynamic unfavored *syn*-diastereomers, which present as the important backbones of many biological active natural products[1–3], remains underdeveloped and desired for a long time[25,26].

The particular chemo- and enantioselective [2 + 2] photocycloadditions have been well established by means of, e.g., dual-catalysis, chiral thioxanthone catalysis, and asymmetric Lewis acid catalysis approaches pioneered by Yoon[17,18], Bach[14,27], and Meggers[19,28] et al. Synergic action of photosensitizer with precise interacting-sites is designed, and the major product of *anti*-diastereomers is often obtained. In case of specific regio- and stereoselectivity of *syn*-diastereomers, the challenge is prominent due to a requirement to maintain *syn*-configuration of two olefins against steric and thermodynamic biases no matter they are the same or different[25,29]. Known examples are mostly based on the solid-state or templated stoichiometric reactions under UV light without catalytic sense, using crystals, zeolites, silica, or hydrogen-bonding templates[29–36]. Very recently, Weiss and coworkers[25] reported a rare example of visible-light catalysis, by precise control of quantum dots (QDs), to achieve tunable selectivity between *syn*-head-to-head (*syn*-HH) and *syn*-head-to-tail (*syn*-HT) diastereomers through [2 + 2] cycloaddition of 4-vinylbenzoic acids, which is assisted by carboxylate group binding to QD surface. The well-organized supramolecular hosts like organic and metal-organic containers (MOCs)[29,37–41] are expected to provide alternative supramolecular confinement effect for [2 + 2] photocycloaddition with anomalous selectivity. Unfortunately, this photocatalytic protocol remains in its infancy, mainly due to drawbacks like absence of visible-light sensitization, (sub) stoichiometric loading of host, and limited scope of substrates. However, considering the rapid progress in supramolecular catalysis of MOCs for many thermal reactions[42–44], we anticipate, by appropriate functionalization of MOCs as supramolecular photoreactors for cage nanospace confinement effect and autocatalytic recycling process, a promising mimic-enzyme photocatalysis of [2 + 2] cycloaddition may be reached as shown in Fig. 1.

To implement MOC nanoreactors as practicable synthetic tool for photocatalysis in solution, several benchmarks may be met as illustrated in Fig. 1a–c: a) visible-light sensitizer incorporation for efficient energy or electron transfer, b) cage-confined nanospace engineering for dynamic substrates inclusion and pre-orientation, and c) reaction acceleration and product release for catalytic turnover. We envisage a supramolecular cage effect with these attributes interplaying in a nanoscale chemical-space, imparting supramolecular confinement to the cage-confined catalysis for

unique reactivity and selectivity[45–47]. Herein, we demonstrate this idea by virtue of a Ru(II)-incorporating metal-organic container (MOC-16) bearing 12 open box-like portals for dynamic guests inclusion and exchange (Fig. 1b, Supplementary Fig. 1, and Tables 1–4)[48]. The $[Pd_6(RuL_3)_8]^{28+}$ nanocage is organized with eight $RuL_3$-metalloligands in a truncated octahedron to form visible-light photoactive nanospace[49], facilitating the confined photocatalysis of intermolecular [2 + 2] cycloaddition with differentiable regio- and diastereoselectivity from the isotopic media[50]. Moreover, water-soluble MOC-16 renders a hydrophobic effect to transfer insoluble substrates and photoproducts through aqueous media to enable an eco-friendly catalytic process, offering an unprecedented approach for general and efficient synthesis of homo- and heterocoupled *syn*-HH diastereomers from chalcones, cinnamates, and benzylideneacetones. To the best of our knowledge, this is the first example to be able to apply a single catalytic-cage to a wide scope of substrates, tackling the problem of balancing the specific selectivity and substrates generalization difficult for enzyme-mimicking catalysis. Moreover, the truly efficient photocatalysis with very low cage-catalyst loading is achieved, solving the problem encountered in cage-catalysis with high selectivity but often product inhibition[42,45].

## Results

**Reaction optimization.** Primary [2 + 2] photocycloaddition in the presence of MOC-16 was tested by chalcone and ethyl 4-bromocinnamate firstly. As shown in Fig. 1d, e, mixing 6 eq (equivalent to MOC-16) substrates with MOC-16 in a DMSO-$d_6$-$D_2O$ solution (1:3, v-v) leads to obvious guest encapsulation as evidenced by the remarkable upfield shift of proton signals of substrates and splitting of host signals[46,48]. After visible-light irradiation (24 W, blue LED, 450 nm) at room temperature in $N_2$ for 3 h, the resonances of substrate protons disappear while those of MOC-16 restore to empty host. $^1H$ NMR identification of the precipitates proves formation of dimerized photoproducts, of which the major *syn*-HH product **1** from chalcone dimerization has been unambiguous characterized by the X-ray single-crystal structural analysis (Fig. 2, Supplementary Fig. 2, and Tables 5–7). Therefore, chalcone was chosen as the benchmark substrate to optimize the reactivity of [2 + 2] cycloaddition by applying MOC-16 as the photocatalyst (see details in Supplementary Table 8). It turns out that solvent media are essential for photocatalytic performance and outcomes. Excellent yield with uncommonly reversed diastereomeric ratio (d.r.) of 1.5:1 (*syn*-HH:*anti*-HH) is observed when using hydrous DMSO-$H_2O$ mixed solvent, of which a ratio of 1:3 performs best for the photoreaction. Decreasing the reaction time from 12 to 3 h and cage-catalyst loading from 2 to 0.5 mol% only slightly reduces the yields, and surprisingly, even at a very low photocatalyst loading of 0.03 mol%, excellent yields (93% NMR-based and 89% isolated) of *syn*-HH major product (1.5:1 d.r.) can be obtained in just 1 h. Such high efficiency and unusual diastereoselective activity endow MOC-16 with uniqueness as a single-component photocatalyst compared to reported systems[7,16,21,29].

**Homocoupling [2 + 2] cycloaddition photocatalysis.** Various chalcone derivatives were tested under the optimized conditions in 1–24 h, showing outstanding outcomes of homocoupling [2 + 2] photocycloaddition (Fig. 2a). It is worthy of note that, the functionalized chalcones all generate *syn*-HH diastereomers as the major photoproducts, most of which show strikingly improved d.r. values or even exclusive formation of *syn*-HH products. Generally, chalcones with diverse *β*-phenyl functional groups results in better d.r. (**2–13**), as compared to those functionalized in *α*-position by different benzoyl groups (**14–17**).

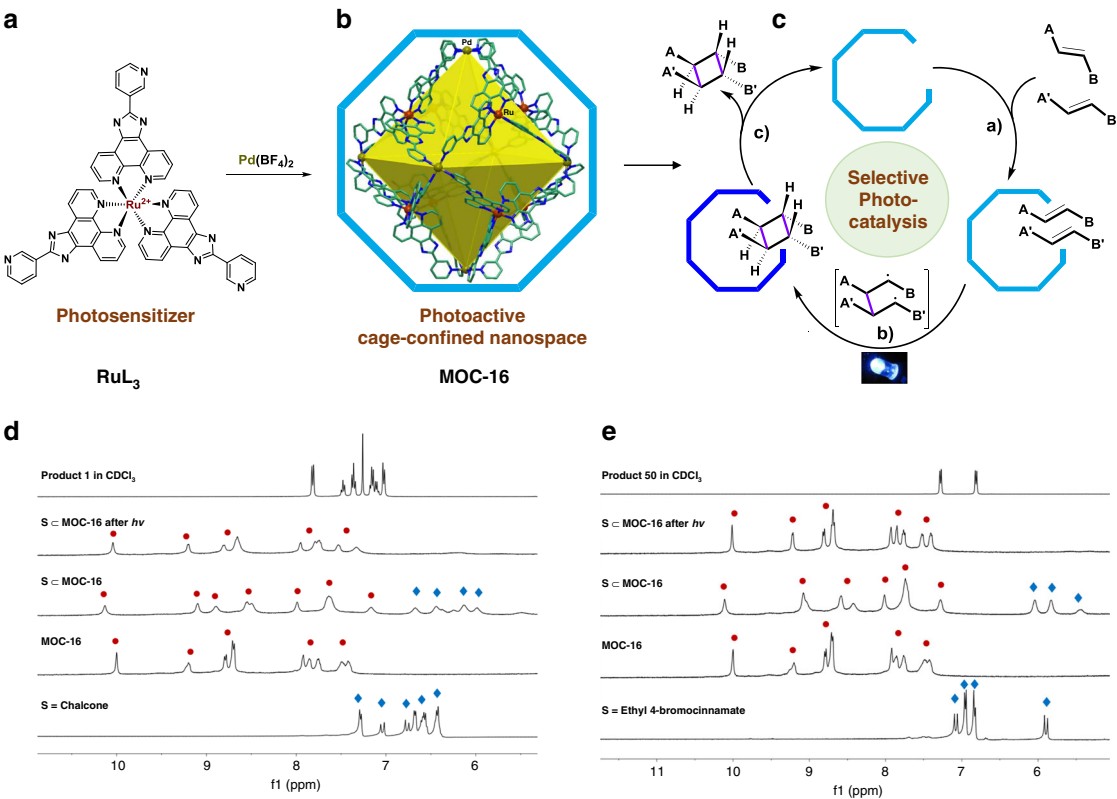

**Fig. 1 Photoactive supramolecular nanoreactor for cage-confined catalysis. a, b** Assembly of nanosized [Pd₆(RuL₃)₈]²⁸⁺ MOC-16 with octahedral cavity organized from RuL₃(BF₄)₂ photosensitizers. **c** Auto-recycled catalytic process enabling visible-light-induced selective [2 + 2] cycloaddition in a cage-confined nanospace to generate *syn*-HH diasteroisomer through: a) encapsulation and pre-orientation of olefins, b) preferable [2 + 2] cycloaddition via efficient triplet energy-transfer, and c) release of cyclobutane product due to host-guest inadaptation. **d** ¹H NMR trace of chalcone photocycloaddition, showing substrate inclusion by MOC-16 (400 MHz, DMSO-*d₆*-D₂O, v-v = 1:3, 298 K), recovery of MOC-16 after product precipitation along LED irradiation, and formation of product **1**. **e** ¹H NMR monitoring of ethyl 4-bromocinnamate photocycloaddition process to form product **50**. Photoproducts are insoluble in mixed DMSO-*d₆*-D₂O solvent and measured in CDCl₃.

Among them, substrates containing *para*-substituted phenyl group (**2–10**) always present excellent diastereoselectivity, most of which give trivial minor *anti*-HH diastereomer that cannot be quantified (**4–6**, **8–10**). Noteworthily, good diastereoselectivity can also be obtained from other functionalized chalcones (**12**, **18**, **19–21**, **27–29**) without limitation to the substitution positions, and some others present relatively low but satisfactory d.r. values in moderate to excellent isolated yields (**11**, **13–17**, **22–26**). Excellent tolerance of diversified functional groups represents another remarkable merit of the photoreaction. Substrates bearing halogen substituents, including chloro (**4**, **25**), bromo (**5**, **11**, **12**, **16**, **17**) and iodo (**6**), which can be further modified readily, show good photocatalytic performance in general. Fluoro-containing functional groups, which are frequently utilized in bioactive compounds and pharmaceuticals, are appropriate substituents for cycloaddition reaction (**3**, **7**, **13**, **15**, **27**, **28**). Substrate with unprotected carboxyl group also offers excellent d.r. value and good isolated yield of product (**8**). In addition, pyridine-, thiophene-, and furan-containing products (**19–24**, **27–29**), which are of interest to medical researchers, are generated smoothly and selectively by applying this photocatalytic protocol. Not only mono-substituted substrates, but also di-substituted ones (**25–29**), are suitable candidates for this photoreaction.

Similar prominent photocatalytic performances are also found applicable to various cinnamates (Fig. 2b). *β*-truxinic ester derivatives, presenting as an important family of natural products[1–3], can be generated facilely in moderate to high yields (**30–61**). Generally, *para*-functionalized cinnamates result in

excellent d.r. and good outcomes regardless of electron-donating or withdrawing groups (**31**, **33–39**, **48–52**); while *ortho* and *meta*-substituted cinnamates show moderate d.r. values (**40**, **42**, **43–46**, **53**) despite exceptions (**41**, **54**), and usually lowered yields with bulky substituents. Though ethyl cinnamates perform similar as compared to methyl cinnamates, introduction of bulkier groups in *α*-position (ester group) influences the behavior of photoreactions more obviously, and sometimes leads to decreased d.r. values, in comparison with functionalization in *β*-position (phenyl group), consistent with observations for chalcone derivatives. Substrates with heterocycle groups are also investigated, resulting in moderate yields and satisfactory to excellent diastereoselectivities (**55–58**). It is noteworthy that *β*-truxinic esters bearing hydroxyl or methoxy group (**33**, **60** and **61**) are obtained exclusively as the *syn*-HH diastereomers. Among them, the dimethoxy *β*-truxinic ester **60** was previously produced in 25%[2] and 40%[3] yields by solid-state photodimerization as the crucial intermediate for caracasandiamide synthesis. Moreover, satisfactory results are achieved when applying *para*-substituted benzylideneacetones as substrates (Fig. 2c), and better diastereoselectivity is observed for the substrates with electron-withdrawing groups (**62–65**).

**Heterocoupling [2 + 2] cycloaddition photocatalysis.** With above promising homodimerization results in hand, we further investigated the cross [2 + 2] photocycloaddition of chalcones, cinnamates, and benzylideneacetones in order to produce

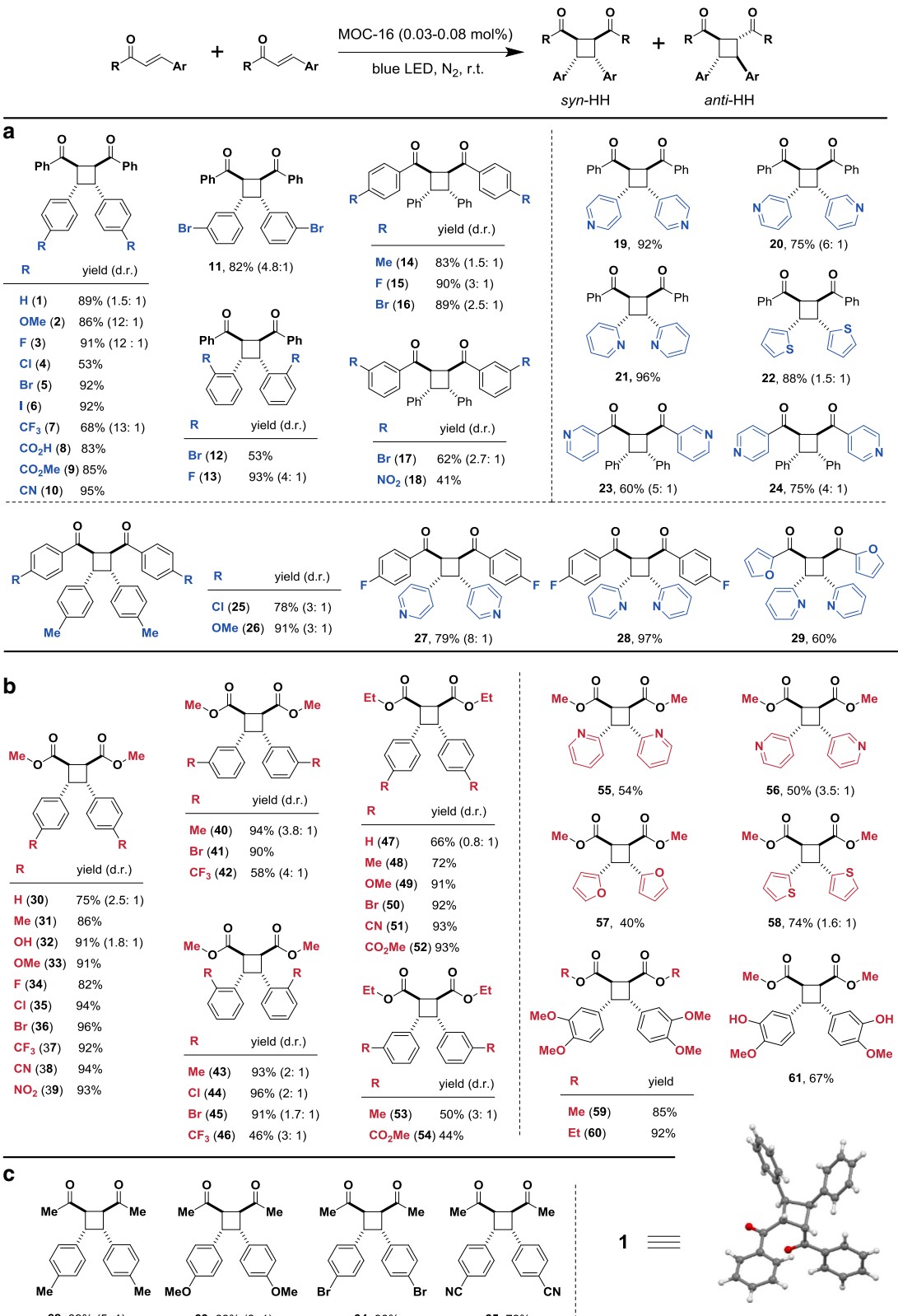

**Fig. 2 Substrates scope of homocoupling [2 + 2] cycloadditions.** Isolated yields are shown with d.r. in parentheses determined by [1]H NMR analyses. No d.r. value means excellent diastereoselectivity for the *syn*-HH isomer with the *anti*-HH isomer undetectable. The data are based on average of two runs. Substrates of **a** chalcone, **b** cinnamate, and **c** benzylideneacetone derivatives. LED light-emitting diode (450 nm); r.t. room temperature. The single-crystal structure of product **1** is demonstrated.

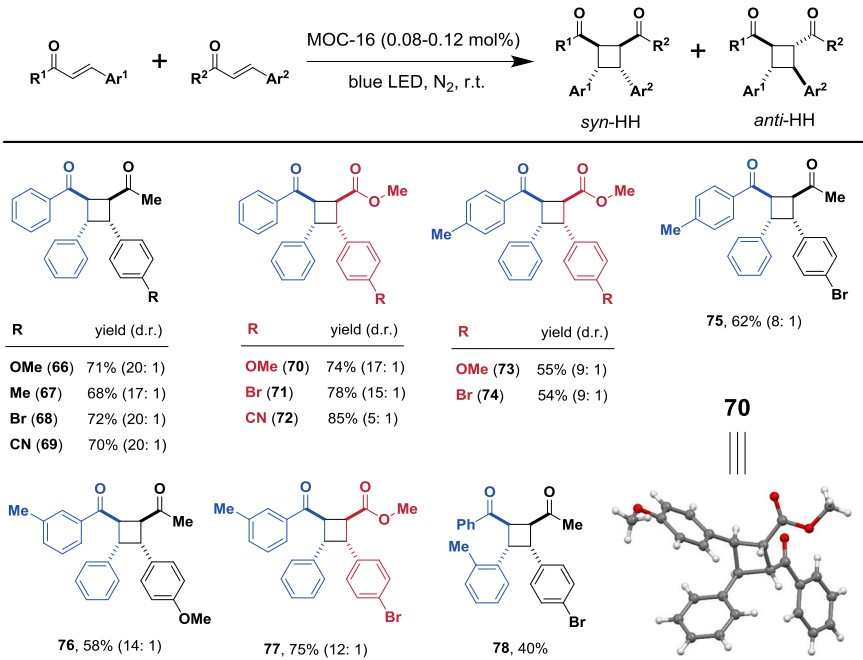

**Fig. 3 Substrates scope of cross [2 + 2] cycloadditions.** Isolated yields are shown with d.r. values in parentheses determined by [1]H NMR analyses. The data are based on average of two runs. The single-crystal structure of product **70** is demonstrated.

asymmetric *syn*-HH diastereomers as major product, which is a formidable challenge and seldom achieved to date[25]. Delightfully, cinnamates and benzylideneacetones react with chalcones successfully to generate corresponding heterocoupled photoproducts (Fig. 3). Keeping chalcones (or cinnamates) as limited substrates, addition of benzylideneacetones (or chalcones) in 2 eq excessive amount leads to facile photocycloaddition to form asymmetric *syn*-HH dimers, of which the major product **70** has been undoubtedly identified with single-crystal structure (Fig. 3 and Supplementary Fig. 2). Moreover, the substituted chalcones containing *para-*, *meta-*, and *ortho*-methyl groups are all suitable reactants, and good substitution tolerance is also observed for the coupled reactants. All heterocoupling [2 + 2] photoreactions display moderate to good yields and good to excellent d.r. values for the required *syn*-HH products (**66–78**).

An essential prerequisite of heterocoupling [2 + 2] photocycloaddition is the co-encapsulation of two different olefins by the cage, which will be further discussed latter. We found that the better selectivity and performance of heterocoupling reactions were obtained when excess amount (≥2 eq.) of one substrate over the other is added. However, which one is required to be in excess will depend on the difference of substrates combination. For example, 2 eq chalcone was needed when combined with cinnamate, but for heterocoupling of chalcone and benzylideneacetone, the latter was required to be in excess. This implies that the heterocoupling is more related with co-encapsulation efficiency of two distinct substrates than their relative dimerizing rates, since chalcone dimerizes faster than both cinnamate and benzylideneacetone. The presence of one substrate in excess than the other may be helpful to balance their co-localization behaviors, considering that the host-guest encapsulation dynamics and phase transition processes of different substrates are understandably different (see discussion later). In addition, the steric and electronic factors are important for the reaction control, which also profoundly influence the co-localization of two different substrates in the cage nanospace. For example, although the heterocoupled products were obtainable from *p*-substituted cinnamates with chalcone (**70–74**), homocoupling

still predominated for *p*-nitrophenyl substituted cinnamate. For chalcones with different *α*-benzoyl groups, *o*-methylbenzoyl functionalized chalcone failed to give heterocoupled product, while *m*-methyl and *p*-methylbenzoyl functionalized chalcones led to desired products (**73–77**). Therefore, effective heterocoupling may be governed by the following collaborative factors that relatively limit substrates scope in comparison with that of homocoupling: (a) molecular nature of reactants suitable for heterocoupling, and (b) co-localization efficiency of two different substrates. The steric and electronic effects play pivotal role in heterocoupling selectivity and outcome, which not only control the heterocoupling reactivity, but also determine the co-encapsulation synergism.

**Scale-up and recycling reactions**. To check the practicability and robustness of MOC-16 photocatalyst, a gram scale reaction was tested for ethyl 4-bromocinnamate substrate, offering 81% isolated yield of **50** without noticeable loss of desired diastereoselectivity (Supplementary Fig. 4). The catalyst recycling reactions by using 1 mol% MOC-16 (for the sake of exercisable recycle handling) verify little decrease of catalytic performance even after running photoreactions for 10 times (Supplementary Table 9 and Fig. 5). A 20 h continuous photoreaction by adding 10 portions of ethyl 4-bromocinnamate reactants in every 2 h without isolation of MOC-16 catalyst was also carried out, which gives an accumulative turnover number (TON) of 910 (Fig. 4a and Supplementary Table 10), evidently confirming the effective catalytic process mediated by MOC-16.

## Discussion

We propose that the photocycloaddition process may rely on the host-guest chemistry, since the cage-confined nanospace can introduce a supramolecular cage effect to exert extrinsic control of photodimerization. The interactions between MOC-16 and substrates were studied by [1]H NMR titration experiments (Supplementary Fig. 6). In contrast to acetone-$H_2O$ or $CH_3CN$-$H_2O$ mixtures containing MOC-16, remarkable proton signal shift and

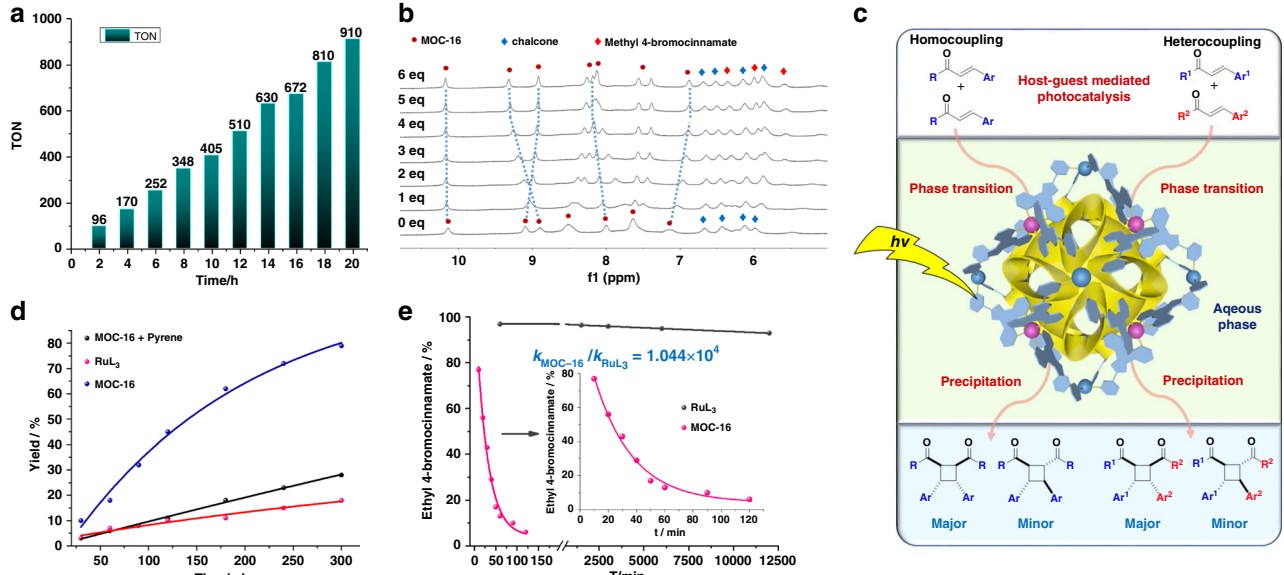

**Fig. 4 Catalytic study. a** Accumulative TONs of production of **50** by 1 mol% MOC-16 with successive addition of 10 portions of substrates in every 2 h without isolation of photocatalyst. **b** $^1$H NMR titration of methyl 4-bromocinnamate into a 5× chalcone ⊂ MOC-16 host-guest system (400 MHz, DMSO-$d_6$: $D_2O$, v-v = 1:3, 298 K) showing co-encapsulation of these two different substrates. **c** Schematic illustration of host-guest mediated photocatalysis of selective homo- and heterocoupling [2 + 2] cycloaddition through an aqueous phase transition expedited process. **d** Inhibition of [2 + 2] dimerization of methyl cinnamate by adding 10 eq competing pyrene guests. **e** Reaction kinetics of ethyl 4-bromocinnamate dimerization catalyzed by 0.08 mol% MOC-16 or 0.64 mol% RuL$_3$ under similar conditions.

split of the host and substrate are observed in DMSO-$d_6$-$D_2O$ (1: 3, v- v) solution with addition of chalcone in aliquots, indicating significant hydrophobic effect in this hydrous medium to facilitate substrate capture[46,48]. Further tests with methyl/ethyl 4-bromocinnamates and 4-bromobenzalacetophenone confirm similar guest encapsulation phenomenon. Addition up to 12 eq substrates does not lead to two sets of independent proton signals belonging to encapsulated and free guests, denoting a dynamic exchange of these substrates[40]. We also notice that, in optimized conditions, all substrates are not well dissolved in hydrous media, giving suspension or emulsion reaction mixtures, and the photoproducts precipitate immediately once formed (Supplementary Fig. 7). This means the water-soluble MOC-16 can solubilize and co-localize hydrophobic reactants via a phase transition process, most probably pre-orienting the substrates in a selective fashion with hydrophobic aryl ends cocooned inside the portals of MOC-16 and hydrophilic carbonyl or ester ends facing hydrous media outside the host[40]. After cycloaddition, the planar substrates transform to unconjugated cyclobutane scaffold with H and substituents turned up to break the host-guest adaptivity, thus released from the host to precipitate.

According to the crystal structure of MOC-16 (Supplementary Fig. 1), the portal has an open box-like cavity of $1.0 \times 1.4$ nm size, which is composed of two parallel L ligands and two bowl-shaped L ligands, thus ready to accommodate and selectively arrange two substrates in one portal, facilitating dimerization and dynamic processes of guest capture and delivery. Since it is hard to grow single-crystals from DMSO-$H_2O$ solution, we tried another way to understand the guest encapsulation behavior of MOC-16. By immersing single-crystals of empty MOC-16 in a n-hexane or n-heptane solution for hours or just minutes, hydrophobic solvent molecules are able to be captured by MOC-16 via a solution-to-solid phase transition process. The single-crystal structure analyses reveal that eight hexane or heptane guests are ordered inside eight portals of MOC-16 (Supplementary Fig. 1c, d). Considering the non-polarity and non-aromatic stacking preference of hexane or heptane, it is rational to find only one guest lying down to

occupy the entire cavity of the portal. These observations suggest strong ability of MOC-16 to sequester and align hydrophobic guests even through a phase transition process, which is assisted by the prominent hydrophobic effect in DMSO-$H_2O$ media.

We further studied the co-encapsulation behavior of MOC-16 for different substrates. As shown in Fig. 4b, we first pre-sequestered five chalcone guests to form a host-guest system of 5× chalcone ⊂ MOC-16 (Supplementary Fig. 6c), and then titrated this system with the second type of substrate of methyl 4-bromocinnamate. Addition of the second guest obviously causes distinct movements of H signals of both host and chalcone, indicating co-localization of these two types of substrates, which is reminiscent of well-known synergism effect in enzyme systems for successive co-encapsulation of guests. Additional encapsulation of five methyl 4-bromocinnamate guests results in a saturated equilibrium with two types of guests adequately accommodated in equal amount by the host, hinting at an origin of heterocoupling cycloaddition owing to co-localization of two different substrates by MOC-16. Above results suggest a viable host-guest mediated photocatalysis of selective homo- and heterocoupling [2 + 2] cycloaddition as depicted in Fig. 4c, where supramolecular cage effect plays a momentous role to capture and pre-orient hydrophobic substrates, control dimerization selectivity, and release the product through a phase transition expedited process.

The mimic-enzyme catalysis sense is clarified by guest-competitive inhibition experiment with a reaction system of pyrene, methyl cinnamate, and MOC-16 (Fig. 4d). Addition of pyrene into the preformed 10× (methyl cinnamate) ⊂ MOC-16 host-guest system results in further encapsulation of pyrene with strong binding propensity[48], and partial release of cinnamate as seen from continuous upfield shift of its proton resonances (Supplementary Fig. 8). The reaction rate is obviously slowed down due to chemical inertness of pyrene, dropping to comparable catalytic performance of RuL$_3$. The reaction kinetics was further investigated for chalcone and methyl cinnamate to compare the catalytic efficiency of MOC-16 and

RuL₃. All photoreactions display apparent first-order behaviors; however, acceleration effect is evidently revealed for the cage-confined catalysis (Supplementary Tables 11–14 and Figs. 9 and 10). For chalcone photodimerization, rate constants are estimated as $k^1$(MOC-16) = 0.0487 min$^{-1}$ and $k^1$(RuL₃) = 0.00415 min$^{-1}$ for MOC-16 and RuL₃ respectively, showing ~12-fold rate enhancement of $k^1$(MOC-16)/$k^1$(RuL₃) = 11.7. For comparison, the cycloaddition of methyl cinnamate exhibits rate constants of $k^2$(MOC-16) = 0.03676 min$^{-1}$ and $k^2$(RuL₃) = 3.5221 × 10$^{-6}$ min$^{-1}$ respectively, giving rise to an extraordinary acceleration effect of $k^2$(MOC-16)/$k^1$(RuL₃) = 1.044 × 10$^4$ (Fig. 4d). Such a high photoreaction enhancement of MOC-16 by four orders of magnitude is striking and valuable, considering that the conventional photocatalysis by RuL₃ is inefficient. Therefore, the supramolecular confinement effect imposed by the cage nanospace evidently predominates the photocatalytic diastereoselectivity and performance of [2 + 2] cycloaddition. Meanwhile, the reaction media and steric and electronic effects also provide additional influence[51] as we mentioned above (Supplementary Table 8).

In photophysical mechanism, the photosensitization is believed to proceed through a triplet energy-transfer rather than an electron-transfer course, since the high reduction potential of reactants (e.g., −1.48 V vs. SCE for chalcone[16] and −1.79 V vs. SCE for ethyl cinnamate[21]) relative to excited *MOC-16 (−0.95 V vs. SCE)[50] eliminates the possibility of electron-transfer procedure, consistent with reported works[13,16,21]. Moreover, in the presence of electron sacrificing trimethylamine or radical trapping TEMPO, photocycloaddition of chalcone shows sharply decreased yields (Supplementary Table 8), further supporting the triplet sensitization mechanism via a diradical intermediate[7,16,18]. To further prove this mechanism, potential E-Z isomerization of olefin was studied since triplet intermediates are known to be able to show trans-cis isomerization. When applying a cinnamate substrate with significantly steric hindrance, namely, ethyl (E)-2-methyl-3-phenylacrylate (Supplementary Fig. 11), its Z-isomer (E/Z ratio = 2.17:1) was obtained successfully without cycloaddition product. In addition, UV-Vis absorption and emission spectra of MOC-16 and chalcone were measured. Exciting MOC-16 at 450 nm results in a broad emission band ~610 nm, which belongs to the ³MLCT emission of MOC-16[49] that well covers the triplet energy area of chalcone (ca. 590 nm)[16]. Emission quenching of MOC-16 is clearly observed in the presence of chalcone, and no absorption of chalcone can be observed in the visible region (Supplementary Fig. 12). These results consist with the triplet energy-transfer mechanism for olefin photosensitization.

In summary, we develop a useful synthetic protocol with unusual selectivity for visible-light intermolecular [2 + 2] cycloaddition of acyclic enones and cinnamates based on proper design and engineering of photoactive cage nanospace in a single MOC supramolecular reactor. The cage-confined catalysis facilitates both homo- and heterocoupling photoreaction to produce syn-HH cyclobutanes in moderate to excellent d.r. values and yields. Good substrate scope and functional group tolerance are achieved with quite low photocatalyst loading. The high catalytic efficiency is attributable to supramolecular cage effect which concentrates reactants, controls selectivity, accelerates reaction, promotes energy-transfer, and expedites recycling catalytic procedure via phase transition delivery. We hope this method will open a potential platform for the broadly useful [2 + 2] photocycloaddition of olefins in a green and sustainable way.

## Methods

**General.** ¹H NMR, ¹³C NMR, and ¹⁹F NMR spectra were recorded at 400, 101, and 377 MHz, respectively, using a Bruker AVANCE III 400 (400 MHz) spectrometer. Measurements were done at ambient temperature. ¹H NMR chemical shifts are referenced to the residual hydrogen signals of the deuterated solvent (7.26 ppm for CDCl₃ and 2.50 ppm for DMSO-$d_6$). The ¹³C NMR chemical shifts are referenced to the ¹³C signals of the deuterated solvent (77.16 ppm for CDCl₃ and 39.52 ppm for DMSO-$d_6$). The ¹⁹F NMR chemical shifts are referenced to the ¹⁹F signals of external CFCl₃ (0 ppm). Abbreviations used in the description of NMR data are listed as follows: br, broad; s, singlet; d, doublet; t, triplet; m, multiplet; and bs, broad singlet. HRMS were recorded on Orbitrap Fusion Lumos or Orbitrap LC/MS (Q Exactive). X-ray diffraction data were collected on an Agilent SuperNova X-Ray diffractometer using micro-focus X-ray sources (Cu-Kα, λ = 1.54184 Å). All solvents are reagent grade or better. Commercially available reagents were used without further purification. Product purification was accomplished by flash chromatography using 200–300 mesh silica gel. Photoirradiation was carried out with blue LED (450 nm) at room temperature.

**Synthesis of MOC-16 and guests ⊂ MOC-16.** MOC-16 containing BF₄⁻ and PF₆⁻ anions was prepared following our previously reported procedure[48]. The cages encapsulating hexane or heptane guest molecules in their rhombic portals were obtained by immersing MOC-16 crystals in hexane or heptane solution for several hours, which led to inclusion of hexane or heptane by MOC-16 through a solution-to-solid phase transfer. The single-crystal structural analyses reveal that each MOC-16 encapsulates eight hexane or heptane molecules in the portals of cage (see Supplementary Methods, Fig. 1, and Tables 1–4).

**Crystal growth of products 1 and 70.** Single-crystals suitable for X-ray diffraction analysis were obtained by natural evaporation of photoproducts **1** in MeOH or **70** in isopropanol, respectively, at room temperature for a few days. See structural analysis details in Supplementary Methods, Fig. 2, and Tables 5–7.

**Preparation of α, β-unsaturated ketones.** See synthetic details in Supplementary Methods.

**Photodimerization of α, β-unsaturated carbonyl compounds.** The photocatalytic [2 + 2] cycloaddition reactions were carried out with blue LED (450 nm, see Supplementary Fig. 3) at room temperature. Five types of general dimerization procedures (A–E) are detailed in Supplementary Methods.

**Large scale photocycloaddition.** A 250-mL Schlenk flask was charged with ethyl 4-bromocinnamate S50 (1 g, 3.9 mmol), MOC-16 (35 mg, 0.08 mol%), and DMSO (30 mL). The mixture was degassed by N₂ for 20 min. Then degassed H₂O (90 mL) was added. The Schlenk flask was then brought into ice bath and irradiated with a 40-W blue LED light strip (see Supplementary Fig. 4) for 24 h. Upon completion, the residue was extracted with EtOAc. The combined organic phase was washed with brine, dried over Na₂SO₄, and evaporated to remove the solvent. The residue was purified by flash chromatography on silica gel (eluting with hexane/ethyl acetate = 100:1) to afford the desired product as a white solid (0.81 g, 81%).

**Recycling experiments of MOC-16 photocatalyst.** General procedure A (see Supplementary Methods) was followed using ethyl 4-bromocinnamate as starting material and 1 mol% MOC-16 as photocatalyst for the sake of exercisable recycle. Upon completion of the reaction (2 h), the mixture was extracted with EtOAc. The organic phase was collected for further purification to obtain the product **50**. The aqueous phase was collected to recycle the catalyst, adding appropriate volume of DMSO/H₂O (1:3) to give a 3-mL solution as before. This solution was added to a Schlenk tube, alone with the ethyl 4-bromocinnamate. The mixture was bubbled with N₂ for 10 min and irradiated with 24 W blue LEDs for another 2 h. The same procedure was repeated for nine times (see Supplementary Fig. 5 and Table 9).

**Continuous photoreactions without isolation of MOC-16 photocatalyst.** General procedure A (see Supplementary Methods) was followed using ethyl 4-bromocinnamate as starting material and 1 mol% MOC-16 as photocatalyst. Ten parallel photoreactions under the same conditions were equipped. After 2 h, one of the reactions was stopped and extracted with EtOAc. The combined organic phase was washed with brine, dried over Na₂SO₄, and then concentrated in vacuo. The residue was dissolved in CDCl₃ for ¹H NMR analyst to test the yield of product for the 2-h reaction. The other nine photoreactions were continued by adding the second portion of ethyl 4-bromocinnamate (0.1 mmol each). The reaction mixtures were bubbled with N₂ for 10 min and irradiated with 24 W blue LEDs for the second 2 h. Such procedures were repeated for nine times, giving rise to the yields for the 10 parallel photoreactions that individually run for 2, 4, 6, 8, 10, 12, 14, 16, 18, and 20 h with supplement of 0.1 mmol ethyl 4-bromocinnamate substrates in every 2 h, respectively. Thus obtained yields approximately represent the conversions of substrates added in 10 portions into a continuous photoreaction lasting for 20 h, offering estimations of the accumulative TONs to produce product **50** (see Fig. 4a and Supplementary Table 10).

**Titration experiments**. Typically, in an NMR tube, MOC-16 (5.5 mg, $5.0 \times 10^{-4}$ mol) was dissolved in 0.15 mL DMSO-$d_6$, followed by addition of 0.45 mL $D_2O$. In total, 5 μL guest solution (0.1 M, in DMSO-$d_6$) was added to the NMR tube and $^1H$ NMR spectra was acquired. Similar procedure was repeated to reach a guest: host ratio up to 9–12. Other host-guest chemistry studies have been carried out in a similar way (see Supplementary Fig. 6).

**Inhibition test**. A 25-mL Schlenk flask equipped with a magnetic stir bar was charged with methyl cinnamate (0.10 mmol), 10 eq pyrene (1 mmol), MOC-16 (0.08 mol%), and DMSO (0.75 mL). The solution was degassed by $N_2$ for 10 min. Then degassed $H_2O$ (2.25 mL) was added. The mixture was irradiated with 24 W LEDs. After a certain time, the reaction was stopped and the residue was extracted with EtOAc. The combined organic phase was washed with brine, dried over $Na_2SO_4$, and then concentrated in vacuo. The residue was purified by flash chromatography on silica gel (eluting with hexane/EtOAc = 100:1) to obtain the yield of product for a certain reaction time. Totally seven reactions under similar conditions with different time (up to 5 h) were examined. The photoreactions with MOC-16 and $RuL_3$ as catalyst without competing guest pyrene were performed in the similar way.

The $^1H$ NMR titration for guest competition of pyrene and methyl cinnamate was carried out to compare with the host-guest titrations with pyrene or methyl cinnamate individually. In an NMR tube, 5.5 mg MOC-16 ($5.0 \times 10^{-4}$ mol) and 0.81 mg methyl cinnamate ($5.0 \times 10^{-3}$ mol, 10 eq.) were dissolved in 0.15 mL DMSO-$d_6$, followed by adding 0.45 mL $D_2O$. In all, 5 μL solution of pyrene (1 eq, 0.1 M in DMSO-$d_6$) was added to the NMR tube and the $^1H$ NMR spectrum was acquired. The same procedure was repeated up to addition of 10 eq pyrene (see Supplementary Fig. 8).

**Kinetic study**. The kinetic studies using chalcone or ethyl 4-bromocinnamate as the substrates are detailed in Supplementary Methods, Figs. 9 and 10, and Tables 11–14.

## Data availability

The data supporting the findings of this study are available within the paper and its Supplementary Information.

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

## Acknowledgements

Financial support was provided by NSFC Projects (21821003, 21720102007, and 21890380), the LIRT Project of GPRTP (2017BT01C161), and the FRF for the Central Universities. P.H. thanks Sun Yat-Sen University for funding. This work is dedicated to Professor Wolfgang Kaim on the occasion of his 70th birthday.

## Author contributions

C.Y.S. conceived and coordinated the project. P.H designed and directed the catalysis. J.S.W. carried out catalytic experiments. C.Y.S., P.H., and J.S.W. wrote the manuscript. K.W. and K.L grew the cage crystals. K.W. performed crystal analyses, and C.Y, Y.H, J.R., and X.F. helped in substrate and catalyst syntheses. All authors discussed the results and commented on the manuscript.

## Competing interests

The authors declare no competing interests.
