## [Peer Review file · Nature Communications]

REVIEWER COMMENTS

Reviewer #1 (Remarks to the Author):

Authors describe use of a metal-organic cage that contains a built-in Ru sensitizer for capturing visible light and catalyzing a 2 + 2 cycloaddition of acyclic alkenes to give syn head-to-head cyclobutanes in high yield and with v. good to outstanding diastereoselectivity.

This paper represents, in my opinion, one of the best examples of the use of supramolecular cages to catalyze a challenging photochemical reaction. There is much to like about this chemistry:

- 1) the reaction protocol is simple and straightforward
- 2) the reaction can be carried out reliably on gram scale
- 3) the catalyst is robust, reusable and has good turnover number over multiple reaction cycles.
- 4) The reaction has a wide substrate scope (65 examples of cinnamates and chalcone derivatives were provided in this paper, with good (50-60%) to excellent yields (> 90%).
- 5) Isolation of product is simple, as it precipitates from solution

This is beautiful system and I recommend its publication after some consideration to the following points:

1) Triplet Intermediate. How do the authors know that this process occurs via a triplet vs. energy transfer? I didn't see any experiments that address this point. They seem to argue for the triplet mechanism based on literature precedent vs. any mechanistic experiments. Triplet intermediates should show trans-cis isomerization. Did the authors observe any cis-olefin during early stages of reaction? Or products that might arise from cis-olefins?

2) Cross-Coupling. One missing point is a reasonable explanation for the high efficiency of the cross-coupling reactions. What factors lead to favoring heterocoupling over homocoupling when mixtures of 2 alkenes are irradiated? Selective heterocoupling is challenging to achieve even with a template, because it requires selective co-localization of two different substrates. Does the catalyst govern these steric factors or are they electronic factors that control cross coupling. The authors provide these nice results but I could see no rationale or explanation for these results.

3) The explanation about Figure 4B, describing co-encapsulation, is confusing and not very convincing to this reviewer.

4) Do the changes in chemical shifts in the NMR data in Fig. S6 tell us anything about guest orientation inside the portals of the cage?

Reviewer #2 (Remarks to the Author):

The authors report a well-performed study on the [2+2]-cycloaddition of cinnamates or chalcones, which proceed with very good and - remarkably - opposite stereochemistry that is usually observed. The key to success is a caged photocatalyst that apparently allows the photoreactions to proceed in a confined environment which accounts for the high diastereoselectivity. A very good number of examples demonstrates the credibility of this approach, and moreover, very low catalyst loading is possible, and finally, a gram scale application is also demonstrated. In my opinion these are very exciting results, and I recommend publication in Nat. Commun. of this fine paper.

The following might be considered:

It is known that the stereochemistry of this reaction is very dependent on the reaction media, and the better a putative diradical intermediate is better stabilized, the higher the trans selectivity (for a very recent study see Chem Open 2020, doi: 10.1002/open.202000092). In turn, the less stabilized such an intermediate, the higher the cis selectivity (i.e. electron-deficient substrates). Looking at the examples provided in this study, such an electronic bias seems not to be operating here, nevertheless, the diastereoselectivity under optimized conditions is very different (e.g. Fig 2a: R = H: 1.5:1; R = OMe: 12:1, R = CN: >99:1). Given the relatively similar size of these groups, I wonder if an electronic effect is responsible for the selectivity in addition to the cage confinement proposed. Thus, in order to clearly demonstrate the cage effect, I would recommend that a few more examples are run with the corresponding uncaged Ru-photocatalyst under the reaction conditions. The authors did this for R=H (Fig 2a) as can be found in the SI, but some more control experiment, e.g. with R=OMe or R=CN would be clearly useful.

Reviewer #3 (Remarks to the Author):

This manuscript reports the use of a photocatalytic metal-organic container (MOC-16) to prepare substituted cyclobutanes through light-induced homo- and hetero-[2+2] cycloaddition reactions.

The major claims of this work are that MOC-16 catalyses the cycloaddition reactions to preferentially form syn-HH cyclobutanes in moderate to good yield and that the cage is critical to the photocatalytic process. These claims are very well supported by the experimental data. The authors show generally good diastereomeric ratios favouring the syn diastereomers across a range of substrate classes with various substituents. The critical role of the cage in the catalytic process was supported by convincing solution-phase host-guest studies showing encapsulation of the substrates by ¹H NMR, single crystal analysis of alkane alignment within the portals of the cage, an inhibition experiment whereby pyrene was used to block the putative catalytic sites of the cage, and a control experiment using a mononuclear analogue of one cage vertex and its ligands (which performed poorly as a catalyst). The outcomes of these different experiments clearly converge upon the authors' conclusion that the hydrophobic portals of the cage serve to entrap and thus concentrate and orient the substrates, and facilitate energy transfer from the Ru(II) chromophores.

In addition to a range of [2+2] homocoupling reactions, the authors also impressively demonstrate conditions whereby MOC-16 can produce heterocoupled products—a particular challenge for [2+2] cycloadditions. Additionally, the authors demonstrate that the photocatalytic reaction can be carried out at gram scales with similar catalytic performance to microscale reactions, and can be used to prepare compounds with pharmacological relevance.

Overall, this is a highly interesting, novel and important piece of work that advances the state-of-the-art in cage-catalysed chemical transformations. I am confident this work will appeal to scientists in fields spanning catalysis, organic synthesis and supramolecular chemistry. Particularly important are the insights into the role of portals around the surface of the cage, in which the catalytic effect is proposed to occur. This contrasts with the emphasis typically placed on the central internal cavity in explanations of cage-mediated catalysis, and may help inform future approaches in the rational design of supramolecular catalysts. As a final note, the detail and quality of the supporting information are among the best I have seen. I have no reservations recommending publication of this excellent work.

While the manuscript is generally very good, I can suggest some minor corrections/additions:

- Please could the authors provide more explicit comment as to why the homocoupling reactions are suppressed under these conditions. I presume homocoupling was reduced by supplying the slower-dimerising alkene in excess, thereby increasing the probability of a heterocoupling reaction rather than homocoupling of the faster-dimerising alkene?
- I noticed a few instances where a contraction (e.g., doesn't, can't) was used. I recommend using "cannot" and "do not" instead.
- In the caption for Figure 2, I recommend adding a note to explain that no d.r. value means excellent diastereoselectivity for the syn-HH isomer. While it is indicated in the main text that the syn-HH isomer is preferred, it would be helpful if this was also explicated in the caption.
- On page 7, in the sentence " β -truxinic esters bearing hydroxyl or methoxy group (32, 33 and 60)", there is disagreement between Figure 1 and the text here. The figure shows that compound 32 was obtained with 1.8:1 d.r., whereas the text indicates exclusive syn-HH formation. This is most likely a typographical error, and suggests that the 1.8:1 d.r. refers to another compound. Please check all d.r. entries in the revised manuscript.
- On page 10, the sentence "...by guest-competitive inhibition experiment with a reaction system of pyrene, methyl cinnamate and MOC-16 (Fig. 4c)". Figure 4c does not appear to detail any competitive inhibition experiment.

Signed, DAR

Point-by-point response to the reviewers' comments

To reviewer #1 (Remarks to the Author):

Comment: “Authors describe use of a metal-organic cage that contains a built-in Ru sensitizer for capturing visible light and catalyzing a 2 + 2 cycloaddition of acyclic alkenes to give syn head-to-head cyclobutanes in high yield and with v. good to outstanding diastereoselectivity.

This paper represents, in my opinion, one of the best examples of the use of supramolecular cages to catalyze a challenging photochemical reaction. There is much to like about this chemistry:

- 1) *the reaction protocol is simple and straightforward*
- 2) *the reaction can be carried out reliably on gram scale*
- 3) *the catalyst is robust, reusable and has good turnover number over multiple reaction cycles.*
- 4) *The reaction has a wide substrate scope (65 examples of cinnamates and chalcone derivatives were provided in this paper, with good (50-60%) to excellent yields (> 90%).*
- 5) *Isolation of product is simple, as it precipitates from solution”*

Response: Thank the reviewer for the very positive comments.

Comment: “This is beautiful system and I recommend its publication after some consideration to the following points:

- 1) *Triplet Intermediate. How do the authors know that this process occurs via a triplet vs. energy transfer? I didn't see any experiments that address this point. They seem to argue for the triplet mechanism based on literature precedent vs. any mechanistic experiments. Triplet intermediates should show trans-cis isomerization. Did the authors observe any cis-olefin during early stages of reaction? Or products that might arise from cis-olefins?”*

Response: Thank the reviewer for the good advice. It is true that the Ru-based complexes are well known from literature for the triplet sensitization in photoreactions. Our previous works have reported the visible light absorption and triplet photoluminescence of MOC-16, as well as its photoredox potential of the excited state of *MOC-16. In this paper, we compared the reduction potentials of *MOC-16 and substrates to exclude the possibility of electron-transfer pathway, and carried out the control experiments in the presence of electron sacrificing trimethylamine and radical trapping TEMPO to prove the triplet sensitization mechanism via a diradical intermediate.

As suggested by the reviewer, we have supplemented the following experiment to support the triplet energy-transfer mechanism for olefin photosensitization. Two substrates (chalcone and ethyl 4-bromocinnamate) have been tested and the reaction mixtures have been checked after running the reactions for different times (5, 10, 15, 30, 60 and 180 min, respectively). However, no *cis*-isomer was observed, indicating that these diradical intermediates are active enough for [2+2] cycloaddition vs. *trans-cis* (or *E-Z*) isomerization under the reaction conditions. Alternatively, a cinnamate substrate with significantly steric hindrance (ethyl (*E*)-2-methyl-3-phenylacrylate) was synthesized, and the *Z*-isomer was obtained successfully without cycloaddition product under the reaction conditions, resulting in mixed olefins with an *E/Z* ratio of 2.17: 1.

In addition, UV-Vis absorption and emission tests of MOC-16 and chalcone have been proceeded and shown in the following figures. The results also support the triplet mechanism. Exciting MOC-16 at 450 nm resulted in a broad emission band around 610 nm, which belongs to the $^3\text{MLCT}$ emission of MOC-16 (*Nat. Commun.* **2016**, *7*, 13169) and well covers the triplet energy area of chalcone (*ca.* 590 nm, *Angew. Chem. Int. Ed.* **2017**, *56*, 15407–15410). Emission quenching is clearly observed by exciting the mixed solvent of MOC-16 and chalcone, and no absorption of chalcone can be observed in the visible region. These observations consist with the triplet energy transfer mechanism.

Above experiments and discussion have been added in the manuscript and Supplementary Information.

Supplementary Figure 12. Absorption and emission spectra of MOC-16 and chalcone. (a) The emission spectrum of MOC-16 (black line), and the absorption spectra of MOC-16 (blue line) and chalcone (red line). (b) The emission quenching of MOC-16 by chalcone at 298 K. For (a), (b), excitation wavelength is 450 nm, the solvent is DMSO: H₂O (1: 3), MOC-16= 1×10⁻⁵ M. For (a) [chalcone] = 1.0×10⁻⁴ M

Comment: 2) *Cross-Coupling.* One missing point is a reasonable explanation for the high efficiency of the cross-coupling reactions. What factors lead to favoring heterocoupling over homocoupling when mixtures of 2 alkenes are irradiated? Selective heterocoupling is challenging to achieve even with a template, because it requires selective co-localization of two different substrates. Does the catalyst govern these steric factors or are they electronic factors that control cross coupling. The authors provide these nice results but I could see no rationale or explanation for these results.”

Response: As the reviewer pointed out, selective heterocoupling is challenging to achieve because it requires selective co-localization of two different substrates. We are pleased by the cross-coupling results and tried hard to understand the mechanism. The co-encapsulation behavior of different substrates by MOC-16 have been studied (Figure 4b), although the detailed information of selective

co-localization is not clear enough because the NMR cannot afford well resolved signals of the guests. However, encapsulation of equal amount of 5 chalcone and 5 methyl 4-bromocinnamate to result in a saturated equilibrium with two types of guests adequately accommodated by the host is evident, hinting at an origin of heterocoupling cycloaddition owing to co-localization of two different substrates by MOC-16.

We believe that effective heterocoupling may be governed by the following collaborative factors that relatively limit substrates scope in comparison with that of homocoupling: a) molecular nature of reactants suitable for heterocoupling, and b) co-localization efficiency of two different substrates. The steric and electronic effects play pivotal role in heterocoupling selectivity and outcome, which not only control the heterocoupling reactivity, but also determine the co-encapsulation synergism.

Besides above discussion added in the text, we also supplement the following explanation based on the additional experiments:

“An essential prerequisite of heterocoupling [2+2] photocycloaddition is the co-encapsulation of two different olefins by the cage, which will be further discussed latter. We found that the better selectivity and performance of heterocoupling reactions were obtained when excess amount (≥ 2 eq.) of one substrate over the other is added. However, which one is required to be in excess will depend on the difference of substrates combination. For example, 2 eq chalcone was needed when combined with cinnamate, but for heterocoupling of chalcone and benzylideneacetone, the latter was required to be in excess. This implies that the heterocoupling is more related with co-encapsulation efficiency of two distinct substrates than their relative dimerizing rates, since chalcone dimerizes faster than both cinnamate and benzylideneacetone. The presence of one substrate in excess than the other may be helpful to balance their co-localization behaviors, considering that the host-guest encapsulation dynamics and phase transition processes of different substrates are understandably different (see discussion later). In addition, the steric and electronic factors are important for the reaction control, which also profoundly influence the co-localization of two different substrates in the cage nanospace. For example, although the heterocoupled products were obtainable from *p*-substituted cinnamates with chalcone (**70-74**), homocoupling still predominated for *p*-nitrophenyl substituted cinnamate. For chalcones with different α -benzoyl groups, *o*-methylbenzoyl functionalized chalcone failed to give heterocoupled product, while *m*-methyl and *p*-methylbenzoyl functionalized chalcones led to desired products (**73-77**).”

Comment: 3) *The explanation about Figure 4B, describing co-encapsulation, is confusing and not very convincing to this reviewer.*”

Response: Thank the reviewer for the reminding. We have revised the relevant discussion carefully to make the explanation more clearly. As we mentioned above, the main point about Figure 4b is that MOC-16 can co-encapsulate two different kinds of substrates (chalcone and methyl 4-bromocinnamate), which is the precondition of the selective cross-coupling. To demonstrate the co-encapsulation behavior of two different kinds of substrates by MOC-16, titration of MOC-16 after pre-encapsulating 5 equiv of chalcone was performed with addition of 5 aliquots of methyl 4-bromocinnamate, which shows step-by-step co-encapsulation procedure to understand the interactions both between the host and the guest and between two different guests. It is clear that addition of methyl 4-bromocinnamate obviously causes successive and distinguishable movements

of H signals of both host and chalcone, indicating co-localization of these two types of substrates similar to synergism effect in enzyme systems for guests co-encapsulation. More importantly, additional encapsulation of 5 methyl 4-bromocinnamate guests results in a saturated equilibrium with two types of guests adequately accommodated in equal amount by the host, hinting at an origin of heterocoupling cycloaddition owing to co-localization of two different substrates by MOC-16.

Comment: 4) Do the changes in chemical shifts in the NMR data in Fig. S6 tell us anything about guest orientation inside the portals of the cage?"

Response: Fig. S6 shows the encapsulation performance of substrates by MOC-16 in different solvents. We are reluctant to tell too much about the detailed guest orientation because the NMR cannot afford well resolved signals of the guests, and the relevant relations of different H atoms cannot be identified from the widened and overlapped peaks clearly.

To reviewer #2 (Remarks to the Author):

Comment: *The authors report a well-performed study on the [2+2]-cycloaddition of cinnamates or chalcones, which proceed with very good and - remarkably - opposite stereochemistry that is usually observed. The key to success is a caged photocatalyst that apparently allows the photoreactions to proceed in a confined environment which accounts for the high diastereoselectivity. A very good number of examples demonstrates the credibility of this approach, and moreover, very low catalyst loading is possible, and finally, a gram scale application is also demonstrated. In my opinion these are very exciting results, and I recommend publication in Nat. Commun. of this fine paper.*

Response: We highly appreciate the very positive comments from reviewer.

Comment: *The following might be considered:*

It is known that the stereochemistry of this reaction is very dependent on the reaction media, and the better a putative diradical intermediate is better stabilized, the higher the trans selectivity (for a very recent study see Chem Open 2020, doi: 10.1002/open.202000092). In turn, the less stabilized such an intermediate, the higher the cis selectivity (i.e. electron-deficient substrates). Looking at the examples provided in this study, such an electronic bias seems not to be operating here, nevertheless, the diastereoselectivity under optimized conditions is very different (e.g. Fig 2a: R = H: 1.5:1; R = OMe: 12:1, R = CN: >99:1). Given the relatively similar size of these groups, I wonder if an electronic effect is responsible for the selectivity in addition to the cage confinement proposed. Thus, in order to clearly demonstrate the cage effect, I would recommend that a few more examples are run with the corresponding uncaged Ru-photocatalyst under the reaction conditions. The authors did this for R=H (Fig 2a) as can be found in the SI, but some more control experiment, e.g. with R=OMe or R=CN would be clearly useful."

Response: Thanks the reviewer for the constructive suggestion. We have referenced the mentioned work, and tested more examples using RuL₃ as photocatalyst. The results and the comparisons

between MOC-16 and RuL₃ are listed in the following figures. It is obvious that the electronic and steric effects indeed affect the reactions in certain degree, but the substrates with electron-deficient substitutes do not always show better *syn*-HH selectivity, even with worse d.r. sometimes. For all of the tested examples in this work, applying MOC-16 as photoreactor leads to better yields and d.r. values. Therefore, we think the supramolecular confinement effect imposed by the cage nanopore should predominate the photocatalytic diastereoselectivity and performance of [2+2] cycloaddition. Meanwhile, the reaction media and steric and electronic effects also provide additional influence. These results and discussion have been added in the revised manuscript and SI (Supplementary Table 8).

Reaction conditions: substrate (0.1 mmol), RuL₃ (0.64 mol%) or MOC-16 (0.08 mol%) and DMSO: H₂O (1: 3, 3 mL) were stirred at room temperature (r.t.) under N₂ with irradiation by 24 W blue light-emitting diode (LED) for 10 h (general procedure B). Yields of applying RuL₃ are determined by ¹H NMR using mesitylene as the internal standard. Yields of applying MOC-16 are isolated yields. d.r. values were obtained by ¹H NMR analyses of reaction mixtures. nd, not detected. ^aMixed acetone: H₂O (2: 1) was used as the solvent (general procedure C).

To reviewer #3 (Remarks to the Author):

Comment: *This manuscript reports the use of a photocatalytic metal-organic container (MOC-16) to prepare substituted cyclobutanes through light-induced homo- and hetero-[2+2] cycloaddition reactions.*

*The major claims of this work are that MOC-16 catalyses the cycloaddition reactions to preferentially form *syn*-HH cyclobutanes in moderate to good yield and that the cage is critical to the photocatalytic process. These claims are very well supported by the experimental data. The authors show generally good diastereomeric ratios favouring the *syn* diastereomers across a range of substrate classes with various substituents. The critical role of the cage in the catalytic process*

was supported by convincing solution-phase host-guest studies showing encapsulation of the substrates by ^1H NMR, single crystal analysis of alkane alignment within the portals of the cage, an inhibition experiment whereby pyrene was used to block the putative catalytic sites of the cage, and a control experiment using a mononuclear analogue of one cage vertex and its ligands (which performed poorly as a catalyst). The outcomes of these different experiments clearly converge upon the authors' conclusion that the hydrophobic portals of the cage serve to entrap and thus concentrate and orient the substrates, and facilitate energy transfer from the Ru(II) chromophores.

In addition to a range of [2+2] homocoupling reactions, the authors also impressively demonstrate conditions whereby MOC-16 can produce heterocoupled products—a particular challenge for [2+2] cycloadditions. Additionally, the authors demonstrate that the photocatalytic reaction can be carried out at gram scales with similar catalytic performance to microscale reactions, and can be used to prepare compounds with pharmacological relevance.

Overall, this is a highly interesting, novel and important piece of work that advances the state-of-the-art in cage-catalysed chemical transformations. I am confident this work will appeal to scientists in fields spanning catalysis, organic synthesis and supramolecular chemistry. Particularly important are the insights into the role of portals around the surface of the cage, in which the catalytic effect is proposed to occur. This contrasts with the emphasis typically placed on the central internal cavity in explanations of cage-mediated catalysis, and may help inform future approaches in the rational design of supramolecular catalysts. As a final note, the detail and quality of the supporting information are among the best I have seen. I have no reservations recommending publication of this excellent work.”

Response: We highly appreciate the very positive comments from the reviewer.

Comment: While the manuscript is generally very good, I can suggest some minor corrections/additions:

- Please could the authors provide more explicit comment as to why the homocoupling reactions are suppressed under these conditions. I presume homocoupling was reduced by supplying the slower-dimerising alkene in excess, thereby increasing the probability of a heterocoupling reaction rather than homocoupling of the faster-dimerising alkene?

Response: Thank the reviewer for the informative advice. In free reaction conditions without cage confinement, it is reasonable that homocoupling will be reduced by supplying the slower-dimerising alkene in excess, thereby increasing the probability of a heterocoupling reaction rather than homocoupling of the faster-dimerising alkene. In our cases, an essential prerequisite of heterocoupling [2+2] photocycloaddition is the co-encapsulation of two different olefins by the cage. In addition, we believe the reaction media, and the steric and electronic factors are also important for the selectivity and performance control, as we respond in detail to the comment of Reviewer 1#.

Comment: • I noticed a few instances where a contraction (e.g., doesn't, can't) was used. I recommend using “cannot” and “do not” instead.

Response: As suggested, the contractions have been changed and marked in yellow in the revised manuscript.

Comment: • *In the caption for Figure 2, I recommend adding a note to explain that no d.r. value means excellent diastereoselectivity for the syn-HH isomer. While it is indicated in the main text that the syn-HH isomer is preferred, it would be helpful if this was also explicated in the caption.*

Response: Thank the reviewer for the good suggestion. A sentence “No d.r. value means excellent diastereoselectivity for the syn-HH isomer with the anti-HH isomer undetectable” has been added and marked in yellow in the revised caption of Figure 2.

Comment: • *On page 7, in the sentence “ β -truxinic esters bearing hydroxyl or methoxy group (32, 33 and 60)”, there is disagreement between Figure 1 and the text here. The figure shows that compound 32 was obtained with 1.8:1 d.r., whereas the text indicates exclusive syn-HH formation. This is most likely a typographical error, and suggests that the 1.8:1 d.r. refers to another compound. Please check all d.r. entries in the revised manuscript.*

Response: Thank the reviewer for the kind reminding. It is a clerical error. “(32, 33 and 60)” has been changed to “(33, 60 and 61)” and marked in yellow in the revised version. Values of d.r. have been double checked and no typographical error was found.

Comment: • *On page 10, the sentence “...by guest-competitive inhibition experiment with a reaction system of pyrene, methyl cinnamate and MOC-16 (Fig. 4c)”. Figure 4c does not appear to detail any competitive inhibition experiment.*

Response: Thank the reviewer for the suggestion. It is a typographical error. “(Fig. 4c)” has been changed to “(Fig. 4d)” and marked in yellow.

REVIEWERS' COMMENTS:

Reviewer #1 (Remarks to the Author):

I reviewed this paper earlier and was very positive about this work but I had 2 concerns about 1) evidence for a triplet intermediate and 2) rationalization for why the cross-coupling [2 + 2] product dominated. I have now read the point by point responses to my concerns (and those of the other reviewers) and also the revised manuscript. The authors have done an excellent job in responding and I am satisfied. I recommend publication of this version of the research.

Reviewer #2 (Remarks to the Author):

I already liked the paper (and recommended publication) in the first version of this manuscript. The additional experiments performed with the uncaged Ru complexes now further highlight the validity of the approach here. This is excellent work, and - again - I recommend publication.

Reviewer #3 (Remarks to the Author):

In addition to my original comments on the manuscript, I believe the authors have satisfactorily raised the issues raised by myself and the two other reviewers. I recommend publication of the excellent work in Nat. Commun.